# FedRolex: Model-Heterogeneous Federated Learning with Rolling Sub-Model Extraction

**Samiul Alam**[1,2]**, Luyang Liu**[3]**, Ming Yan**[4,1]**, Mi Zhang**[2,1]
[1]Michigan State University, [2]The Ohio State University,
[3]Google Research,    [4]The Chinese University of Hong Kong, Shenzhen
`alamsami@msu.edu, luyangliu@google.com, yanming@cuhk.edu.cn, mizhang.1@osu.edu`

## Abstract

Most cross-device federated learning (FL) studies focus on the model-homogeneous setting where the global server model and local client models are identical. However, such constraint not only excludes low-end clients who would otherwise make unique contributions to model training but also restrains clients from training large models due to on-device resource bottlenecks. In this work, we propose `FedRolex`, a partial training (PT)-based approach that enables model-heterogeneous FL and can train a global server model larger than the largest client model. At its core, `FedRolex` employs a rolling sub-model extraction scheme that allows different parts of the global server model to be evenly trained, which mitigates the client drift induced by the inconsistency between individual client models and server model architectures. We show that `FedRolex` outperforms state-of-the-art PT-based model-heterogeneous FL methods (e.g. Federated Dropout) and reduces the gap between model-heterogeneous and model-homogeneous FL, especially under the large-model large-dataset regime. In addition, we provide theoretical statistical analysis on its advantage over Federated Dropout and evaluate `FedRolex` on an emulated real-world device distribution to show that `FedRolex` can enhance the inclusiveness of FL and boost the performance of low-end devices that would otherwise not benefit from FL. Our code is available at: https://github.com/AIoT-MLSys-Lab/FedRolex.

## 1 Introduction

Federated learning (FL) is a machine learning paradigm that trains models from distributed clients with private data under the coordination of a central server [1, 2]. In this work, we focus on cross-device FL where clients are usually resource-constrained edge devices. The majority of existing cross-device FL studies focus on the *model-homogeneous* setting [3–6], in which the server model and the client models across all the participating client devices are *identical*. However, model-homogeneous FL are confronted with two fundamental constraints: (1) **device heterogeneity** is a more realistic consideration when deploying FL systems in real-world applications: different client devices could have very diverse on-device resources and are only capable of training models with capacities that match their on-device resources. Having the same model on all the devices would, unfortunately, exclude clients with low-end devices who would otherwise make unique contributions to model training from their own local data; (2) state-of-the-art machine learning has moved towards **large models** [7] such as Transformer [8]. Restricting server and client models to be the same inevitably causes model-homogeneous FL to fail to train such large models due to the resource constraint of client devices.

To relax the fundamental constraints of model-homogeneous FL, **model-heterogeneous FL** was proposed where heterogeneous models with different capacities across the server and the clients are trained during the federated training process. One primary challenge in model-heterogeneous FL is

36th Conference on Neural Information Processing Systems (NeurIPS 2022).

Table 1: Comparison of `FedRolex` with model-homogeneous and model-heterogeneous FL methods.

| | Model Heterogeneity | Aggregation Scheme | Sub-model Extraction Scheme | Need of Public Data | Server Model Size | Compatibility with Secure Aggregation |
|---|---|---|---|---|---|---|
| FedAvg [3] | | | | No | = Client Model | Yes |
| FedProx [4] | No | - | - | No | = Client Model | Yes |
| SCAFFOLD [5] | | | | No | = Client Model | Yes |
| FedBE [6] | | | | Unlabeled | = Client Model | No |
| FedGKT [9] | | | | No | ≥ Largest Client Model | No |
| FedDF [10] | Yes | Knowledge Distillation | - | Unlabeled | = Largest Client Model | No |
| DS-FL [11] | | | | Unlabeled | = Largest Client Model | No |
| Fed-ET [12] | | | | Unlabeled | ≥ Largest Client Model | No |
| Federated Dropout [13] | | | Random | No | ≥ Largest Client Model | Yes |
| HeteroFL [14] | Yes | Partial Training | Static | No | = Largest Client Model | Yes |
| FjORD [15] | | | Static | No | = Largest Client Model | Yes |
| **FedRolex (Our Approach)** | | | **Rolling** | **No** | **≥ Largest Client Model** | **Yes** |

the aggregation of heterogeneous client models. To address this challenge, knowledge distillation (KD)-based approaches have been proposed [9–12], in which the client models serve as teachers and the server ensembles the knowledge distilled from the individual client models. However, KD-based approaches in general require public data on the server to achieve competitive model accuracy, whereas the desired public data may not be always available in practice. Moreover, since KD-based approaches need the individual client models (whole models or prediction layers) or their outputs to be sent to the server, they are incompatible with secure aggregation protocols [16], which limits their privacy guarantee. To remove the dependency on public data and ensure compatibility with secure aggregation, partial training (PT)-based approaches such as random sub-model extraction (Federated Dropout [13]) and static sub-model extraction (HeteroFL [14], FjORD [15]) were proposed. In these approaches, each client trains a smaller sub-model extracted from the larger global server model, and the server model is updated by aggregating those trained sub-models. However, the fundamental issue of existing PT-based methods is that the sub-models are extracted in ways (either random or static) such that the parameters of the global server model are *not evenly trained*. This makes the server model vulnerable to client drift[1] induced by the *inconsistency between individual client model and server model architectures* – a unique challenge of model-heterogeneous FL.

In this work, we propose a PT-based model-heterogeneous FL approach named `FedRolex` to tackle the fundamental issue of existing methods. The key difference between `FedRolex` and existing PT-based methods is how the sub-models are extracted for each client over communication rounds in the federated training process. Specifically, instead of extracting sub-models in either random or static manner, `FedRolex` proposes a **rolling sub-model extraction** scheme, where the sub-model is extracted from the global server model using a rolling window that advances in each communication round. Since the window is rolling, sub-models from different parts of the global model are extracted in sequence in different rounds. As a result, all the parameters of the global server model are *evenly trained* over the local data of client devices.

The proposed rolling sub-model extraction scheme, though simple, has equipped `FedRolex` with multifold merits compared to prior arts (Table 1): (1) `FedRolex` enables different parts of the global server model to be evenly trained, which mitigates the client drift induced by model heterogeneity. (2) Contrary to static sub-model extraction approaches (HeteroFL, FjORD), `FedRolex` is able to train a global server model that is *larger* than the largest client model, enabling FL to benefit from the superior performance brought by large models. It echoes some concurrent efforts in developing FL primitives to support training large server models in cross-device settings, e.g. Federated Select [17]. (3) Compared to random sub-model extraction (Federated Dropout), as we show in our theoretical statistical analysis in Section 3 and Appendix A.1, the global server model is trained more evenly by `FedRolex` as the expected number of rounds for `FedRolex` going through all the parameters of the global model for at least certain times is smaller than that of Federated Dropout. (4) `FedRolex` only needs to transmit the sub-model that is needed by a given client instead of the full server model to the client. This allows clients to contribute to federated training under resource constraints and reduces communication overheads (Appendix A.6). (5) Lastly, `FedRolex` is fully compatible with existing secure aggregation protocols that enhance the privacy properties of FL systems.

We evaluate the performance of `FedRolex` under two regimes: i) small-model small-dataset regime (most existing cross-device FL studies use this combination), and ii) large-model large-dataset regime

---

[1]In model-homogeneous FL, client drift is primarily induced by *data heterogeneity* across clients. In model-heterogeneous FL, *model heterogeneity* across clients is *another* critical source that induces client drift.

(this combination echos recent efforts on pushing the frontier of cross-device FL towards training large server models on large-scale datasets [18–21]). We highlight five of our findings: (1) `FedRolex` consistently outperforms state-of-the-art PT-based model-heterogeneous FL methods under both small-model small-dataset and large-model large-dataset regimes (§4.1). (2) `FedRolex` reduces the gap between model-heterogeneous and model-homogeneous FL, especially under large-model large-dataset regime (§4.2). (3) With `FedRolex`, under both regimes, having a small fraction of large-capacity models could significantly boost the global model accuracy (§4.3). (4) `FedRolex` is able to train a global server model that is larger than the largest client model and outperforms Federated Dropout in terms of global model accuracy (§4.4). (5) Using an emulated real-world device distribution, we show that `FedRolex` enhances the **inclusiveness** of FL and boosts the performance of low-end devices that would otherwise not benefit from FL (§4.5).

## 2    Related Work

**Knowledge Distillation (KD)-based Model-Heterogeneous FL.** One primary approach for model-heterogeneous FL in cross-device settings is based on knowledge distillation (KD) [22]. In particular, FedDF [10] distills knowledge from a set of classifiers trained with private data from a federation of client devices. The logit outputs of each classifier against an unlabeled public dataset are then used to train a student model at the server with KD. Similarly, DS-FL [11] utilized an unlabeled public dataset at the server and proposed a distillation-based semi-supervised FL approach to enhance performance by pseudo-labeling the public data. FedGKT [9] proposed group knowledge transfer in which knowledge is transferred to a large model in the server from clients without public data. Fed-ET [12] proposed a weighted consensus distillation scheme with diversity regularization that enables the training of a large server model with smaller client models. KD-based approaches, however, have several limitations: they often require public data to achieve competitive model accuracy. This is because model accuracy is dependent on the size of public data as well as the domain similarity of public data with client data [10, 12, 23]. Furthermore, as KD-based methods use client model weights partially or entirely as teachers to transfer knowledge to the server, they are incompatible with secure aggregation protocols, making them vulnerable to backdoor attacks [20].

**Partial Training (PT)-based Model-Heterogeneous FL.** To address the limitations of KD-based approaches, partial training (PT) has emerged as another solution for model-heterogeneous FL. Depending on how the sub-models are extracted from the global server model, existing PT-based methods can be in general categorized into two groups: *random* sub-model extraction and *static* sub-model extraction. Specifically, inspired by the dropout technique commonly used in centralized training [24], Federated Dropout [13] proposed to randomly extract sub-models from the global model. Though easy to be integrated into existing FL frameworks, as reported in [25], Federated Dropout becomes less effective when the data heterogeneity is high and the client cohort is small due to its randomness in selecting sub-models. In contrast, HeteroFL [14] and FjORD [15] proposed static extraction schemes where sub-models are *always* extracted from a *designated* part of the global server model. However, such a static extraction strategy has two primary drawbacks. First, the global server model is restricted to the *same* size as the largest client model. As such, the size and capability of the global model are implicitly restricted by the resources of client devices, making it not able to train large models due to resource bottlenecks at client devices. Second and more importantly, under static extraction, depending on their resource demands, different sub-models can *only* be trained on clients whose on-device resources are matched. As a consequence, part of the global server model cannot be trained on data at low-end client devices, causing different parts of the global model to be trained on data with different distributions. This would degrade the performance of the global model, especially under high data heterogeneity. In this work, we propose a rolling sub-model extraction scheme that tackles the issues of both random and static sub-model extraction methods.

## 3    Methodology

### 3.1    Formulation of Model-Heterogeneous FL

Let $\mathcal{N}$ denote $N$ client devices with non-IID (non-identically and independently distributed) local data $D = \{D_1, D_2, ..., D_N\}$. Model-homogeneous FL trains a global model of parameter $\theta$ by solving the following optimization problem:

$$\min_{\theta} F(\theta) \triangleq \sum_{n=1}^{N} p_n F_n(\theta) \tag{1}$$

with

$$F_n(\theta) \triangleq \frac{1}{m_n} \sum_{k=1}^{m_n} l(\theta; d_{n,k}), \tag{2}$$

where $D_n \triangleq \{d_{n,1}, d_{n,2}, d_{n,3}...d_{n,m_n}\}$ is the set of local data samples of client $n$ and $p_n$ is its corresponding weight such that $p_n \geq 0$ and $\sum_{n=1}^{N} p_n = 1$.

In comparison, in *model-heterogeneous* FL, clients train local models with heterogeneous capacities $\boldsymbol{\beta} = \{\beta_1, \beta_2, ..., \beta_N\}$, and the local objective function of the $n^{th}$ client becomes

$$F_n'(\theta_n) \triangleq \frac{1}{m_n} \sum_{k=1}^{m_n} l(\theta_n; d_{n,k}). \tag{3}$$

Here, $\beta_n$ denote the model capacity of client $n$, and we define it as the proportion of nodes extracted from each layer in $\theta$ for client $n$. The size of $\theta_n$ depends on $\beta_n$, and the parameter $\theta_n$ is obtained by selecting a sub-model from the global model $\theta$, which can change from one round to another. If $\theta_n$ changes, the objective function also changes. For simplicity, we use the same notation $l$ for the loss function for all clients and rounds, though they differ between clients and rounds. The key to model-heterogeneous FL is selecting $\theta_n$ from the global model $\theta$ given model capacity $\beta_n$.

### 3.2 FedRolex: Model-Heterogeneous FL with Rolling Sub-Model Extraction

As a partial training (PT)-based approach, at each client, `FedRolex` trains only a sub-model extracted from the global server model and sends the corresponding sub-model updates back to the server for update aggregation. To help understand how `FedRolex` works, for simplicity, Figure 1 illustrates three rounds of federated training of `FedRolex` on two participating heterogeneous clients, where one trains a large-capacity sub-model (left) and the other trains a small-capacity one (right). At the high level, at each round, the server extracts sub-models of different capacities from the global model and separately broadcasts them to the clients that have the corresponding capabilities. The clients train the received sub-models on their local data and transmit their heterogeneous sub-model updates to the server. Lastly, the server aggregates those updates, and the result of the aggregation is used to update the global model for the next round. The pseudocode of `FedRolex` is in Algorithm 1.

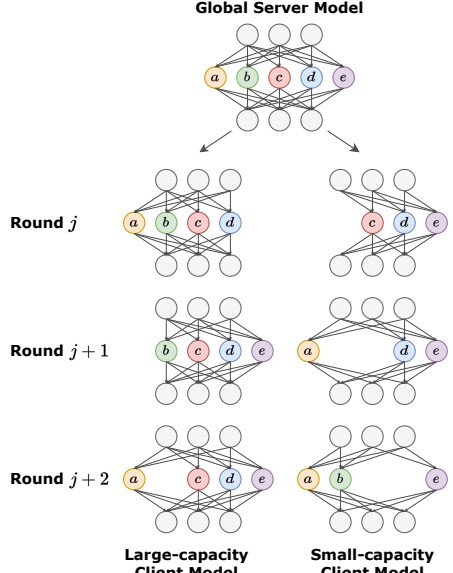

Figure 1: Overview of the rolling sub-model extraction scheme in `FedRolex`.

The key to the design of `FedRolex` involves two design choices. In the following, we describe them in detail.

**(1) What sub-models to be extracted for each client across different rounds?** At the server, `FedRolex` utilizes a rolling window to extract the sub-model from the global model. The rolling window advances in each round, and loops over all parts of the global model *in sequence* across different rounds. This process iterates such that the global model is evenly trained until convergence.

Taking Figure 1 as an example: in round $j$, the large-capacity and small-capacity client model extracted from the global model is $\{a, b, c, d\}$ and $\{c, d, e\}$, respectively. In round $j + 1$, the rolling window advances one step[2], the large-capacity and small-capacity client model becomes $\{b, c, d, e\}$ and $\{d, e, a\}$, respectively. Similarly, in round $j + 2$, the rolling window advances one step further, and the large-capacity and small-capacity client model becomes $\{c, d, e, a\}$ and $\{e, a, b\}$, respectively.

Such a rolling sub-model extraction scheme can be formalized as follows. Let $\theta_n^{(j)}$ denote the parameters of the sub-model extracted from the global model for client $n$ in round $j$, $K_i$ denote the

---

[2]The step size is a hyperparameter of `FedRolex`. Please refer to Appendix A.4 for our ablation study on it.

total number of nodes in layer $i$ of the global model, and $\mathcal{S}_{n,i}^{(j)}$ denote the node indices of layer $i$ of the global model that belongs to the extracted sub-model for client $n$ in round $j$. Then the layer $i$ of the sub-model extracted by the rolling sub-model extraction scheme for client $n$ in round $j$ is given by:

$$\mathcal{S}_{n,i}^{(j)} = \begin{cases} \{\hat{j}, \hat{j}+1, \ldots, \hat{j}+\lfloor \beta_n K_i \rfloor - 1\} & \text{if } \hat{j} + \lfloor \beta_n K_i \rfloor \leq K_i, \\ \{\hat{j}, \hat{j}+1, \ldots, K_i - 1\} \cup \{0, 1, \ldots, \hat{j}+\lfloor \beta_n K_i \rfloor - 1 - K_i\} & \text{else.} \end{cases} \tag{4}$$

where $\hat{j} = j \bmod K_i$.

**(2) How to aggregate heterogeneous sub-model updates to update the global model?** `FedRolex` employs a straightforward selective averaging scheme with no client weighting to aggregate heterogeneous sub-model updates sent from the clients to update the global model[3]. Specifically, it computes the average of the updates for each parameter of the global model separately based on how many clients in a round updated that parameter. The parameter remains unchanged if no clients updated it.

Taking Figure 1 again as an example: in round $j$, the updates for $a$ and $b$ are obtained from the large-capacity model and the update for $e$ is from the small-capacity model only. In contrast, since $c$ and $d$ are part of both models, the update is computed by taking the average from both models.

---

**Algorithm 1: FedRolex**

---

1 Initialization ; $\theta^{(0)}, \mathcal{N}$
    **Input** : $D_n$ $\beta_n$ $\forall n \in \mathcal{N}$,
    **Output** : $\theta^J$
2 Server Executes
3 **for** $j \leftarrow 0$ **to** $J - 1$ **do**
4     Sample subset $\mathcal{M}$ from $\mathcal{N}$
5     Broadcast $\theta_{m,\mathcal{S}_{m,i}^{(j)}}^{(j)}$ to client $m \in \mathcal{M}$
6     $\forall i, \mathcal{S}_{m,i}^{(j)}$ from Equation (4)
7     **for** *each client* $m \in \mathcal{M}$ **do**
8         `clientStep`$(\theta_m^{(j)}, D_m)$
9     **end**
10     Aggregate $\theta_{[i,k]}^{(j+1)}$ according to Equation (10)
11 **end**

12 **Subroutine** `clientStep`$(\theta_n^{(j)}, D_n)$
13     $m_n \longleftarrow len(D_n)$
14     **for** $k \leftarrow 0$ **to** $m_n$ **do**
15         $\theta_n \longleftarrow \theta_n - \eta \nabla l(\theta_n; d_{n,k})$
16     **end**
17     return $\theta_n$

---

### 3.3 Comparison with Random and Static Sub-model Extraction Schemes

Existing sub-model extraction schemes can be grouped as random-based (Federated Dropout) and static-based (HeteroFL, FjORD) methods. In this section, we describe the differences between them and the proposed rolling-based scheme employed in `FedRolex`. For comparison purpose, the pseudocodes of both Federated Dropout and HeteroFL are included in Appendix A.8.

#### 3.3.1 Comparison with Random Sub-Model Extraction Scheme

In random sub-model extraction scheme, in each round, the sub-models are extracted from the global model in a random manner. As such, the layer $i$ of the sub-model extracted by the random sub-model extraction scheme for client $n$ in round $j$ is given by:

$$\mathcal{S}_{n,i}^{(j)} = \{k_c \mid \text{integer } k_c \in [0, K_i - 1] \text{ for } 1 \leq c \leq \lfloor \beta_n K_i \rfloor\}, \tag{5}$$

where a total number of $\lfloor \beta_n K_i \rfloor$ nodes are randomly chosen from the global model.

**Discussion:** As shown in Figure 2(left), similar to the proposed rolling-based scheme, the sub-models extracted across different rounds by the random-based scheme have different architectures. However,

---

[3]In Appendix A.3, we did an ablation study on three client weighting schemes and compared them with the non-weighting (selective averaging) scheme. We find that the performance of the three weighting schemes is not significantly better than the non-weighting scheme. Please refer to Appendix A.3 for details.

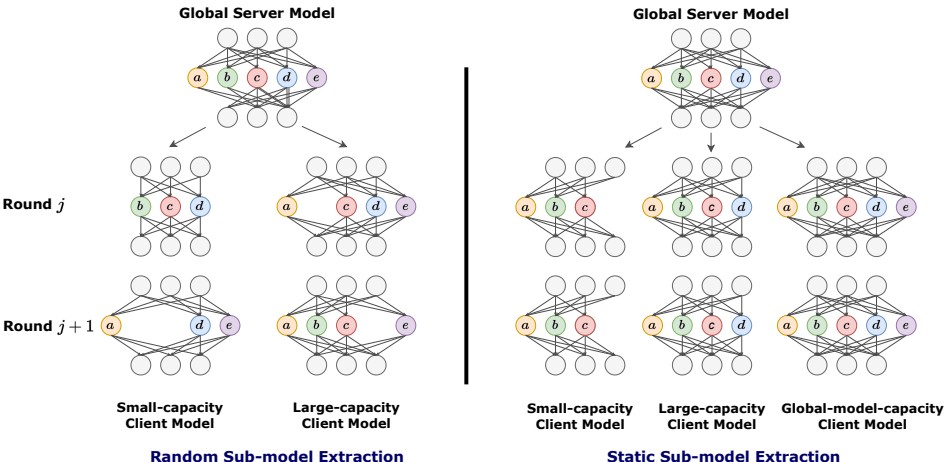

Figure 2: Illustration of how sub-models are extracted by random sub-model extraction scheme (Left) and static sub-model extraction scheme (Right) over two rounds.

due to its randomness in selecting sub-models in each round, the global model is trained less evenly, making it more vulnerable to client drift. In short, although the expected value of the frequency for updating each index is the same for all the indices, their exact frequencies are not the same due to randomness. Consequently, the random-based scheme cannot balance the update frequencies of different parts of the global model, and it inevitably takes more rounds to update the whole global model. Moreover, as we show in Appendix A.1, the expected number of rounds for Federated Dropout selecting all $I$ sub-models at least $m$ times is in the order of $I\log(I) + I(m-1)\log\log I$, which is larger than that of `FedRolex`, $mI$.

### 3.3.2 Comparison with Static Sub-Model Extraction Scheme

In static sub-model extraction scheme, in each round, the sub-models are *always* extracted from a *designated* part of the global model. As such, the layer $i$ of the sub-model extracted by the static sub-model extraction scheme for client $n$ in round $j$ is given by:

$$\mathcal{S}_{n,i}^{(j)} = \{0, 1, 2, \ldots, \lfloor \beta_n K_i \rfloor - 1\}. \tag{6}$$

Note that $\mathcal{S}_{n,i}^{(j)}$ does *not* depend on $j$. In other words, as shown in Figure 2(right), the *same* sub-model is extracted for each client in *every* round. Moreover, the client model with smaller capacity and client model with larger capacity are *not independent*. As shown in Figure 2(right), the small-capacity model $\{a, b, c\}$ is a *part* of the large-capacity model $\{a, b, c, d\}$, which again, is a *part* of the global-capacity model $\{a, b, c, d, e\}$. These are the two key differences between both the random-based and the proposed rolling-based scheme.

**Discussion:** Given that, the static-based scheme, however, has two primary drawbacks. First, to cover the whole global model, there must be clients to train the full-size global model $\{a, b, c, d, e\}$. As such, the global model is restricted to the *same* size as the largest client model. Second, as shown in Figure 2(right), while $a$, $b$ and $c$ will be trained on data on all three types of clients, $d$ will not be trained on data on small-capacity clients, and $e$ will only be trained on data on global-model-capacity clients. As a consequence, different parts of the global model are trained on data with different distributions, which inevitably degrades the global model training quality.

## 4 Experiments

**Datasets and Models.** We evaluate the performance of `FedRolex` under two regimes. Under small-model small-dataset regime, we train pre-activated ResNet18 (PreResNet18) models [26] on CIFAR-10 and CIFAR-100 [27]. We replace the batch Normalization in PreResNet18 with static batch normalization [14, 28] and add a scalar module after each convolution layer [14]. Under large-model large-dataset regime, we use Stack Overflow [29] and followed [2] to train a modified 3-layer Transformer [8] with a vocabulary of $10,000$ words, where the dimension of token embeddings is

128, and the hidden dimension of the feed-forward network (FFN) block is 2048. We use ReLU activation and use 8 heads for the multi-head attention where each head is based on 12-dimensional (query, key, value) vectors. The statistics of the datasets are listed in Table 2.

Table 2: Dataset statistics.

| Dataset | Train Clients | Train Examples | Validation Clients | Validation Examples | Test Clients | Test Examples |
|---|---|---|---|---|---|---|
| CIFAR-10 | 100 | 50,000 | N/A | N/A | N/A | 10,000 |
| CIFAR-100 | 100 | 50,000 | N/A | N/A | N/A | 10,000 |
| Stack Overflow | 342,477 | 135,818,730 | 38,758 | 16,491,230 | 204,088 | 16,586,035 |

**Data Heterogeneity.** For CIFAR-10 and CIFAR-100, we followed HeteroFL [14] to model non-IID distributions by restricting each client to have $L$ labels. In our evaluation, we consider two levels of data heterogeneity. For CIFAR-10, we define $L = 2$ as high data heterogeneity and $L = 5$ as low data heterogeneity. For CIFAR-100, we use $L = 20$ as high data heterogeneity and $L = 50$ as low data heterogeneity. These two levels roughly correspond to Dirichlet distribution $Dir_K(\alpha)$ with $\alpha$ equal to $0.1$ and $0.5$, respectively. For Stack Overflow, the dataset is partitioned over user IDs, making the dataset naturally non-IID distributed.

**Model Heterogeneity.** Without loss of generality, in our evaluation, we consider five different client model capacities $\beta = \{1, 1/2, 1/4, 1/8, 1/16\}$ where for instance, $1/2$ means the client model capacity is half of the largest client model capacity (full model). To generate these client models, for ResNet18, we vary the number of kernels in convolution layers and keep the nodes in the output layers the same. For Transformer, we vary the number of nodes in the hidden layer of the attention heads.

**Baselines.** We compare `FedRolex` against both state-of-the-art PT-based model-heterogeneous FL methods including Federated Dropout [13] and HeteroFL [14][4] as well as state-of-the-art KD-based model-heterogeneous FL methods including FedDF [10], DS-FL [11] and Fed-ET [12] [5]. To ensure a fair comparison, all the PT-based baselines are trained using the same learning rate, number of communication rounds, and multi-step learning rate decay schedule. The details of the schedule for each dataset and experiment are described in Appendix A.7.

**Configurations and Platform.** For CIFAR-10 and CIFAR-100, we apply bounding box crop [30] to augment the images. In each communication round, $10\%$ of the clients are randomly selected from a pool of 100 clients. For Stack Overflow, we followed [2] to use a $10\%$ dropout rate to prevent over-fitting, and 200 clients are randomly selected from a pool of $342,477$ clients in each communication round. The details of the hyper-parameters for model training are included in Appendix A.7. We implemented `FedRolex` and PT-based baselines using PyTorch [31] and Ray [32], and conducted our experiments on 8 NVIDIA A6000 GPUs.

**Evaluation Metrics.** We use global and local model accuracy as our evaluation metrics. Specifically, global model accuracy is defined as the server model accuracy on the test set; and local model accuracy is defined as the accuracy of the server model on each of the client's local datasets. For CIFAR-10 and CIFAR-100, we report the classification accuracy. For Stack Overflow, we report the next word prediction accuracy which includes both out-of-vocabulary (OOV) and end-of-sentence (EOS) tokens. We run our experiments using five different seeds for CIFAR-10 and CIFAR-100 and using three different seeds for Stack Overflow.

## 4.1 Performance Comparison with State-of-the-Art Model-Heterogeneous FL Methods

First, we compare the performance of `FedRolex` with state-of-the-art PT and KD-based model-heterogeneous FL methods. For a fair comparison, we followed the experimental settings used in prior arts where the distributions of client model capacities are uniform and the global server model is the same as the largest client model.

**Evaluation Results:** Table 3 summarizes our results. We have two observations. (1) In comparison with state-of-the-art PT-based methods, under the small-model small-dataset regime, `FedRolex` consistently outperforms HeteroFL and Federated Dropout under both low and more challenging high data heterogeneity scenarios. In particular, under high data heterogeneity, Federated Dropout

---

[4]We did not compare with FjORD because its code is not open-source and we could not reproduce their results following the paper.

[5]We did not compare with FedGKT [9] as it is only compatible with CNN models.

Table 3: Global model accuracy comparison between `FedRolex`, PT and KD-based model-heterogeneous FL methods, and model-homogeneous FL methods. Note that the results of KD-based methods were obtained from [12]. For Stack Overflow, since KD-based methods cannot be directly used for language modeling tasks, their results are marked as N/A.

| Method | | High Data Heterogeneity | | Low Data Heterogeneity | | Stack Overflow |
| | | CIFAR-10 | CIFAR-100 | CIFAR-10 | CIFAR-100 | |
| --- | --- | --- | --- | --- | --- | --- |
| KD-based | FedDF | 73.81 (± 0.42) | 31.87 (± 0.46) | 76.55 (± 0.32) | 37.87 (± 0.31) | N/A |
| | DS-FL | 65.27 (± 0.53) | 29.12 (± 0.51) | 68.44 (± 0.47) | 33.56 (± 0.55) | N/A |
| | Fed-ET | **78.66 (± 0.31)** | **35.78 (± 0.45)** | **81.13 (± 0.28)** | **41.58 (± 0.36)** | N/A |
| PT-based | HeteroFL | 63.90 (± 2.74) | 52.38 (± 0.80) | 73.19 (± 1.71) | 57.44 (± 0.42) | 27.21 (± 0.22) |
| | Federated Dropout | 46.64 (± 3.05) | 45.07 (± 0.07) | 76.20 (± 2.53) | 46.40 (± 0.21) | 23.46 (± 0.12) |
| | `FedRolex` | **69.44 (± 1.50)** | **56.57 (± 0.15)** | **84.45 (± 0.36)** | **58.73 (± 0.33)** | **29.22 (± 0.24)** |
| | Homogeneous (smallest) | 38.82 (± 0.88) | 12.69 (± 0.50) | 46.86 (± 0.54) | 19.70 (± 0.34) | 27.32 (± 0.12) |
| | Homogeneous (largest) | **75.74 (± 0.42)** | **60.89 (± 0.60)** | **84.48 (± 0.58)** | **62.51 (± 0.20)** | **29.79 (± 0.32)** |

which extracts sub-model randomly has worse performance than `FedRolex` and HeteroFL which both extract sub-models in a deterministic manner. Under large-model large-dataset regime, `FedRolex` also outperforms both HeteroFL and Federated Dropout. These results together demonstrate the superiority of `FedRolex` under both regimes. (2) In comparison with state-of-the-art KD-based methods, `FedRolex` only performs worse than Fed-ET and FedDF on CIFAR-10 under high data heterogeneity, but outperforms all the KD-based methods on the more challenging CIFAR-100 which has a larger number classes than CIFAR-10 under both low and high data heterogeneity scenarios. It is important to note that KD-based methods leverage public data to boost their model accuracy while `FedRolex` does not.

## 4.2 Performance Comparison with Model-Homogeneous FL Methods

We also compare the global model accuracy of `FedRolex` with two model-homogeneous cases where all the clients have the largest capacity model ($\beta = \{1\}$) and the smallest capacity model ($\beta = \{1/16\}$), representing the upper and lower-bound performance, respectively.

**Evaluation Results:** As listed in Table 3, compared with other PT-based methods, `FedRolex` reduces the gap in global model accuracy between model-heterogeneous and upper-bound model-homogeneous settings. In particular, `FedRolex` is on par with the upper-bound model-homogeneous case for Stack Overflow, whereas both HeteroFL and Federated Dropout perform even worse than the model homogeneous case using the smallest model. This result indicates that with `FedRolex`, we will not be constrained to only using high-end devices to achieve competitive global model accuracy. Note that Fed-ET achieves a higher global model accuracy than the model-homogeneous upper bound on CIFAR-10 under high data heterogeneity, which showcases the advantage of using public data.

## 4.3 Impact of Client Model Heterogeneity Distribution

In our previous experiments, the distributions of model capacities across client devices are set to be uniform. In this experiment, we aim to understand the impact of the client model heterogeneity distribution. To do so, without loss of generality, we use two client model capacities $\beta = \{1, 1/16\}$ and vary the distribution ratio between the two (denoted as $\rho$) where $\rho = 1$ represents the case in which all the clients have the largest capacity model ($\beta = \{1\}$) and $\rho = 0$ represents the case in which all the clients have the smallest capacity model ($\beta = \{1/16\}$).

**Evaluation Results:** Figure 3 shows how global model accuracy changes when $\rho$ varies from 0 to 1 for CIFAR-10, CIFAR-100 and Stack Overflow. We have three observations. (1) For CIFAR-10 (Figure 3(i)), there is a large gap in global model accuracy between high and low data heterogeneity for a wide range of $\rho$ (from 0.1 to 1). This is because CIFAR-10 is a relatively simple task and hence the global model accuracy is bottlenecked by the level of data heterogeneity instead of model capacity. This result indicates that having more high-capacity models in the cohort has only limited contribution to global model accuracy. (2) For the more challenging CIFAR-100 (Figure 3(ii)), the gap in global model accuracy is much lower between high and low data heterogeneity. In contrast to CIFAR-10, the global model accuracy is bottlenecked by the highest capacity of the models rather than the level of data heterogeneity. (3) For both regimes (Figure 3(i)(ii) vs. Figure 3(iii)), we observe

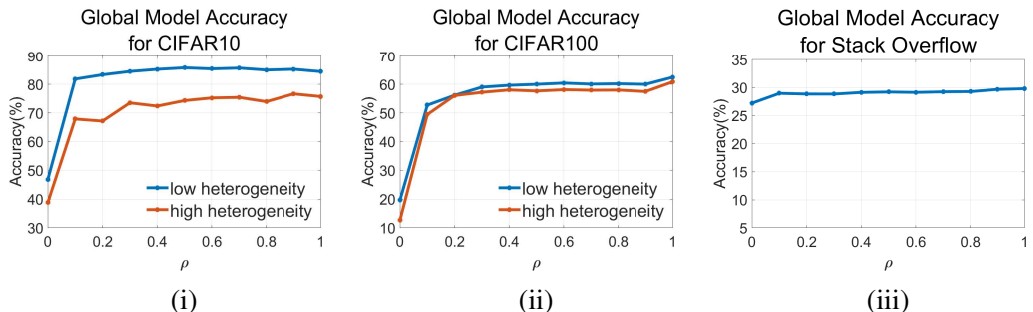

Figure 3: Impact of client model heterogeneity distribution on global model accuracy for (i) CIFAR-10, (ii) CIFAR-100, and (iii) Stack Overflow.

that having a small fraction of large-capacity models significantly boosts the global model accuracy, but keeping increasing the ratio of large-capacity models has limited contribution to the accuracy.

## 4.4 Performance on Training Larger Server Model

Similar to Federated Dropout, one advantage of `FedRolex` over static sub-model extraction methods (HeteroFL and FjORD) is that `FedRolex` is able to train a global server model that is larger than the largest client model. In this experiment, we aim to evaluate the performance of `FedRolex` on training larger server models. To do so, we consider the case where the size of the global server model is $\gamma = \{2, 4, 8, 16\}$ times the size of client models. For simplicity, all client models have the same size.

**Evaluation Results:** Figure 4(i) and Figure 4(ii) compare `FedRolex` with Federated Dropout in terms of global model accuracy when $\gamma$ for CIFAR-10 and CIFAR-100, respectively. As shown, although the global model accuracy drops for both `FedRolex` and Federated Dropout when $\gamma$ increases, especially from 1 to 4, `FedRolex` consistently achieves higher global model accuracy than Federated Dropout across $\gamma = \{2, 4, 8, 16\}$ under both low and high data heterogeneity. For Stack Overflow (Figure 4(iii)), the global model accuracy has a much smaller drop when $\gamma$ increases. This demonstrates the superiority of using large models on large-scale datasets for training larger server models.

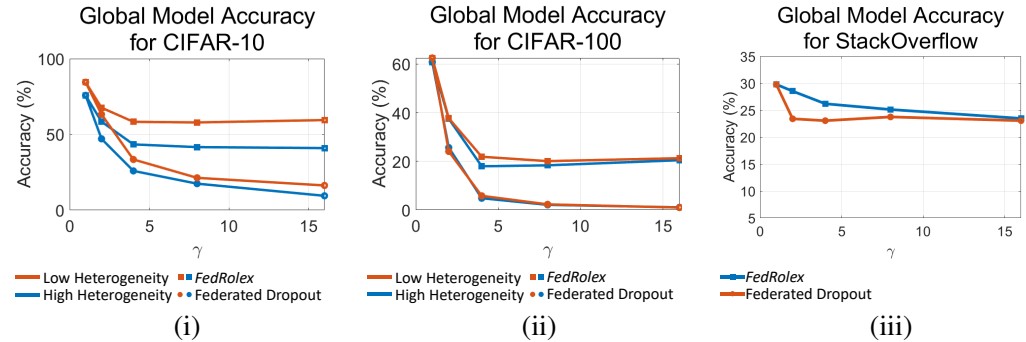

Figure 4: Performance on training larger server model when the server model is $\gamma$ times the size of the client model for (i) CIFAR-10, (ii) CIFAR-100, and (iii) Stack Overflow.

## 4.5 Enhance Inclusiveness of FL in Real-world Distribution

A primary vision of `FedRolex` is to enhance the inclusiveness of FL. To demonstrate this, in this experiment, we use real-world household income distribution to emulate real-world device distribution. Specifically, we retrieve household income distribution information from Bureau [33]. We map $\beta_n = 1/16$ with the income group with earning less than \$75,000 and assign proportions of remaining groups in \$25,000 increments with increasing values of $\beta_n$. Detailed mapping of this distribution to the corresponding income distribution is provided in Figure 7 in Appendix A.7.

Table 4: Performance of FedRolex under emulated real-world device distribution.

| Dataset | Method | High Data Heterogeneity | | Low Data Heterogeneity | |
|---|---|---|---|---|---|
| | | Local Accuracy | Global Accuracy | Local Accuracy | Global Accuracy |
| CIFAR-10 | Homogeneous (smallest) | 85.90 (± 0.46) | 38.82 (± 0.88) | 66.02 (± 0.52) | 46.86 (± 0.54) |
| | Homogeneous (largest) | 95.54 (± 0.26) | 75.74 (± 0.41) | 93.54 (± 0.44) | 84.48 (± 0.58) |
| | FedRolex | 94.05 (± 1.01) | 63.17 (± 1.45) | 91.03 (± 0.36) | 80.14 ± 0.52) |
| CIFAR-100 | Homogeneous (smallest) | 34.51 (± 0.56) | 12.69 (± 0.50) | 33.22 (± 0.10) | 19.70 (± 0.34) |
| | Homogeneous (largest) | 81.99 (± 0.78) | 60.89 (± 0.60) | 76.43 (± 0.54) | 62.51 (± 0.20) |
| | FedRolex | 73.33 (± 0.96) | 45.78 (± 1.71) | 66.31 (± 0.34) | 48.44 (± 0.51) |
| Stack Overflow | Homogeneous (smallest) | 27.32 (± 0.12) | | | |
| | Homogeneous (largest) | 29.79 (± 0.32) | | | |
| | FedRolex | 29.55 (± 0.41) | | | |

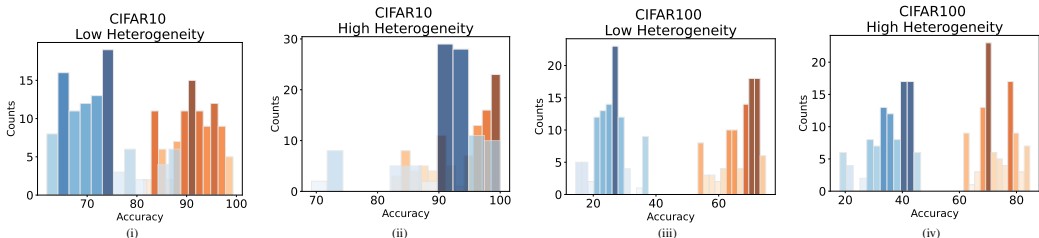

Figure 5: Local model accuracy distribution of `FedRolex` (orange color) vs. the smallest model-homogeneous case (blue color) for CIFAR-10 and CIFAR-100 under low and high data heterogeneity.

**Evaluation Results:** Table 4 shows both the global and local model accuracies of `FedRolex` for CIFAR-10 and CIFAR-100 as well as the global model accuracy on Stack Overflow under the emulated real-world device distribution. Again, we compare with two model-homogeneous cases where all clients have the smallest and largest model capacities, representing lower and upper-bound accuracy, respectively. We make two observations. (1) Looking at the global model accuracy, `FedRolex` consistently outperforms the lower-bound model-homogeneous case across CIFAR-10, CIFAR-100, and Stack Overflow. This result indicates that `FedRolex` enhances the inclusiveness of FL and improves the accuracy of the global model, which would otherwise not be able to achieve. (2) Looking at the local model accuracy, `FedRolex` significantly outperforms the lower-bound model-homogeneous case on CIFAR-10 and CIFAR-100 under both low and high data heterogeneity. This result indicates that `FedRolex` effectively boosts the performance of low-end devices, which would otherwise not benefit from FL. A detailed illustration of how local model accuracy distribution of individual clients shifts when `FedRolex` is used compared to the smallest model-homogeneous case with the same client outreach is shown in Figure 5.

## 5 Conclusion

We presented `FedRolex`, a partial training (PT)-based model-heterogeneous FL approach that is able to train a global server model larger than the largest client model. `FedRolex` proposed a rolling sub-model extraction scheme that enables parameters of the global server model to be evenly trained to mitigate client drift induced by model heterogeneity. We provided a theoretical statistical analysis of its advantage over Federated Dropout. Our experimental results show that `FedRolex` consistently outperforms state-of-the-art PT-based methods across models and datasets at both small and large scales. Moreover, we demonstrated its performance on an emulated real-world device distribution and show `FedRolex` contributes to making FL more inclusive.

**Limitations and Future works.** In this work, we provided a statistical analysis of `FedRolex`. Full convergence analysis of `FedRolex` is not trivial and is left for future work. In addition, the goal of this work is to train a global server model using a federation of heterogeneous client models. Determining what models to deploy onto each client after the global server model is trained is a separate task, especially when the global server model is large. We will pursue it as our future work.

## 6 Acknowledgement

We thank the reviewers for their helpful comments. This work was partially supported by NSF PFI:BIC-1632051, CNS-1814551, DMS-2012439, and a Google Computing Platform (GCP) grant.

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
