## A  Appendix

### A.1  Statistical Analysis

**Lemma 1.** *Given $I$ indices, and one index is chosen at each round equally randomly. The expected number of rounds of choosing all indices at least once is*

$$I\left(\frac{1}{I} + \frac{1}{I-1} + \cdots + \frac{1}{1}\right),$$

*which is the same as*

$$I\int_0^\infty \left(1 - (1 - e^{-t})^I\right) dt.$$

*Proof.* We denote the expected number of rounds to choose exactly $i$ indices at least once as $E(i)$. Then we have $E(1) = 1$, because, after the first round, one index is chosen. After the first round, the expected number of rounds to choose a new index is $\frac{I}{I-1}$, because one of the remaining $I-1$ out of the total $I$ indices needs to be chosen. That is, $E(2) = E(1) + \frac{I}{I-1}$. Similarly, we have

$$E(i) = E(i-1) + \frac{I}{I+1-i}, \qquad \forall i = 2, \ldots, I.$$

Thus, we have

$$E(I) = E(I-1) + I = E(I-2) + \frac{I}{2} + \frac{I}{1} = \cdots = I\left(\frac{1}{I} + \frac{1}{I-1} + \cdots + \frac{1}{1}\right).$$

The lemma is proved. □

It shows that the expected number of rounds to choose all indices at least once is $I\log(I)$ when $I \to \infty$. This proof can not be generalized to the case for choosing all indices at least $m$ times for $m \geq 2$. Therefore, we provide alternative proof for it [34, Example 5.17].

*Alternative proof of Lemma 1.* This proof considers picking the indices as Poisson processes. Assume that the Poisson process to choose one index has a rate $\lambda = 1$. Since the index is chosen equally randomly, choosing the $j$th index also follows a Poisson process with a rate $1/I$ for any $j$ [34, Proposition 5.2]. We let $X_j$ be the time to choose the first index $j$, and

$$X = \max_{1 \leq j \leq I} X_j \tag{7}$$

is the time all indices are chosen at least once. Since all $X_j$ are independent with rate $1/I$, we have

$$P\{X < t\} = P\{\max_{1 \leq j \leq I} X_j < t\} = P\{X_j < t, \text{ for } j = 1, \ldots, I\}$$
$$= (1 - e^{-t/I})^I.$$

Therefore, we have

$$\mathbb{E}[X] = \int_0^\infty P\{x > t\}dt = \int_0^\infty \left(1 - (1 - e^{-t/I})^I\right) dt$$

We let $N$ be the number of rounds to choose all indices at least one, and $T_i$ be the $i$th interarrival time of the Poisson process for choosing one index. Then we have

$$X = \sum_{i=1}^N T_i,$$

and $T_i$ are independent. Thus we have

$$\mathbb{E}[X|N] = N\mathbb{E}[T_i] = N,$$

and which gives

$$\mathbb{E}[X] = \mathbb{E}\{\mathbb{E}[X|N]\} = \mathbb{E}[N].$$

Thus we have

$$\mathbb{E}[N] = \int_0^\infty \left(1 - (1 - e^{-t/I})^I\right) dt = I \int_0^\infty \left(1 - (1 - e^{-t})^I\right) dt.$$

The lemma is proved. $\qquad\qquad\qquad\qquad\qquad\qquad\qquad\qquad\qquad\qquad\qquad\qquad\qquad\qquad\quad\square$

Next, we will present the lemma for choosing each index at least $m$ times.

**Lemma 2.** *Given $I$ indices, and one index is chosen at each round equally randomly. The expected number of rounds of choosing all indices at least $m$ times is*

$$I \int_0^\infty \left(1 - (1 - S_m(t)e^{-t})^I\right) dt,$$

*where*

$$S_m(y) := 1 + y + \frac{y^2}{2!} + \cdots + \frac{y^{m-1}}{(m-1)!} = \sum_{l=0}^{m-1} \frac{y^l}{l!}. \qquad (8)$$

*Proof.* We consider picking the indices as Poisson processes again. Assume that the Poisson process to choose one index has a rate $\lambda = 1$. Since the index is chosen equally randomly, choosing the $j$th index also follows a Poisson process with a rate of $1/I$ for any $j$. We let $X_j$ be the time to choose index $j$ for the $m$th time, and

$$X = \max_{1 \leq j \leq I} X_j \qquad (9)$$

is the time all indices are chosen at least $m$ times. Since all $X_j$ are independent with rate $1/I$, we have

$$P\{X < t\} = P\{\max_{1 \leq j \leq I} X_j < t\} = P\{X_j < t, \text{ for } j = 1, \ldots, I\}$$

$$= (1 - S_m(t/I)e^{-t/I})^I.$$

Therefore, we have

$$\mathbb{E}[X] = \int_0^\infty P\{x > t\} dt.$$

We let $N$ be the number of rounds to choose all indices at least $m$ times, and $T_i$ be the $i$th interarrival time of the Poisson process for choosing one index. Then we have

$$X = \sum_{i=1}^N T_i,$$

and $T_i$ are independent. Thus we have

$$\mathbb{E}[X|N] = N\mathbb{E}[T_i] = N,$$

and which gives

$$\mathbb{E}[X] = \mathbb{E}\{\mathbb{E}[X|N]\} = \mathbb{E}[N].$$

Thus we have

$$\mathbb{E}[N] = \int_0^\infty \left(1 - (1 - S_m(t/I)e^{-t/I})^I\right) dt = I \int_0^\infty \left(1 - (1 - S_m(t)e^{-t})^I\right) dt.$$

The lemma is proved. $\qquad\qquad\qquad\qquad\qquad\qquad\qquad\qquad\qquad\qquad\qquad\qquad\qquad\qquad\quad\square$

It shows that the expected number of rounds to choose all indices at least once is $I \log(I) + I(m - 1) \log \log I$ when $I \to \infty$ [35].

## A.2 Formal Definition of Selective Aggregation Scheme

Formally speaking, let $\mathcal{M} \subset \mathcal{N}$ be the set of selected clients from the client pool from which the server pulls model parameters at round $j$. Let $\theta_{[i,k]}$ be the $k^{th}$ parameter of layer $i$ of the global model and $\theta_{m,[i,k]}$ be the $k^{th}$ parameter of layer $i$ of client $m$. We denote $\mathcal{M}_k \subset \mathcal{M}$ as the set of clients updating the $k^{th}$ parameter. The model parameters are aggregated as follows:

$$\theta_{[i,k]} = \frac{1}{\sum_{m \in \mathcal{M}_k} p_m} \sum_{m \in \mathcal{M}_k} p_m \theta_{m,[i,k]}, \tag{10}$$

The client weight $p_m$ is assigned based on factors like the client model capacity, the number of data points a client has, etc. Throughout the paper, unless otherwise stated, the weight of all clients is assumed to be the same, i.e, $p_m = 1/N$.

## A.3 Ablation Study: Impact of Different Weighing Schemes

[12] reported that weighting clients is important to improving model accuracy. Therefore, we did an ablation study and evaluated three client weighting schemes: (1) **model size-based weighting scheme**: client weight is proportional to the number of kernels in the model; (2) **model update-based weighting scheme**: client weight is proportional to the number of updates; and (3) **hybrid weighting scheme**: client weight is proportional to both (1) model size and (2) model update.

Table 5 lists the results. As shown, the performance of the three weighting schemes is not significantly better than the non-weighting scheme. Therefore, we used the non-weighting scheme in `FedRolex`.

Table 5: Impact of weighting schemes on model accuracy under high data heterogeneity.

| | Weighting Scheme | Local Model Accuracy | Global Model Accuracy |
|---|---|---|---|
| | Non-Weighting | 95.95 (±0.81) | **69.44 (±1.50)** |
| CIFAR-10 | Model Size-based | 95.98 (±0.67) | 69.09 (±1.42) |
| | Model Update-based | 96.01 (±0.71) | 68.83 (±0.89) |
| | Hybrid | **96.05 (±0.96)** | 68.78 (±0.89) |
| | Non-Weighting | **81.58 (±0.59)** | 56.57 (±0.15) |
| CIFAR-100 | Model Size-based | 81.23 (±1.56) | **56.99 (±0.27)** |
| | Model Update-based | 81.23 (±1.07) | 56.63 (±0.36) |
| | Hybrid | 81.49 (±1.07) | 56.71 (±0.20) |

## A.4 Ablation Study: Impact of Overlapping Kernels

We also studied the impact of overlapping kernels between rounds using ResNet-18 and CIFAR-10/CIFAR-100 as an example. Specifically, we extracted sub-models using a rolling window that

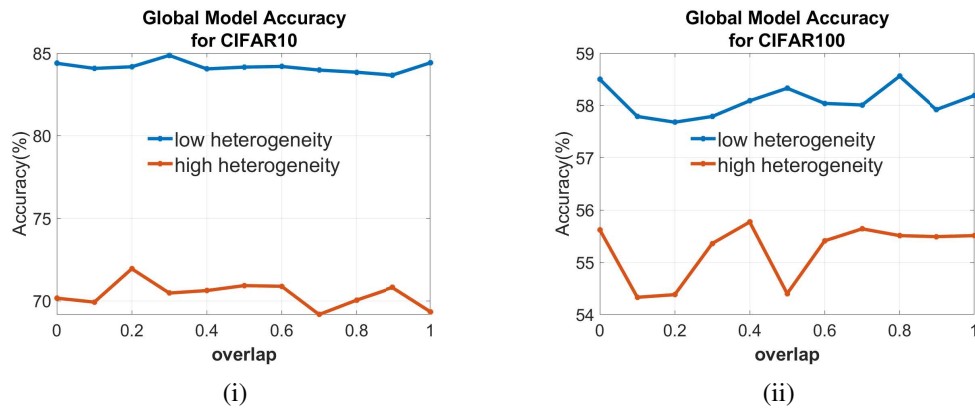

Figure 6: Impact of inter-round kernel overlap on global model accuracy under low and high data heterogeneity for (i) CIFAR-10 and (ii) CIFAR-100.

advances and loops over all the kernels of each convolution layer in the global model in strides. Let the degree of overlap between each stride of the rolling window be $r \in [0, 1]$. In each iteration, each convolution layer in the global model is advanced by $1 + \lfloor \beta_n (1 - r) K_i \rfloor$ where $\lfloor \cdot \rfloor$ is the floor function. In `FedRolex`, $r = 1$, i.e., the kernels are advanced by 1 from one iteration to the next iteration.

Figure 6 shows the impact of different $r$ on global model accuracy. As shown, the value of $r$ does have some influence on the global model accuracy, but the impact is non-linear and inconsistent.

### A.5 Ablation Study: Impact of Client Participation Rate

In our main paper, we followed prior arts [14, 24, 15, 36, 9] and used a 10% client participation rate. To examine the effect of client participation rate, we conducted experiments with both lower (5%) and higher (20%) client participation rates using CIFAR-10 as an example for `FedRolex`, HeteroFL and Federated Dropout.

The results are summarized in Table 6. As shown, `FedRolex` consistently outperforms both Federated Dropout and HeteroFL across 5%, 10% and 20% client participation rates.

Table 6: Performance of FedRolex, HeteroFL, and Federated Dropout under different client participation rates.

| | | Client Participation Rate | | |
|---|---|---|---|---|
| | | 5% | 10% | 20% |
| | HeteroFL | 48.43 (+/- 1.78) | 63.90 (+/-2.74) | 65.07 (+/- 2.17) |
| **CIFAR-10** | Federated Dropout | 42.06 (+/- 1.29) | 46.64 (+/-3.05) | 55.20 (+/- 4.64) |
| | FedRolex | **57.90 (+/- 2.72)** | **69.44 (+/-1.50)** | **71.85 ( +/- 1.22)** |

### A.6 Communication and Computation Costs of FedRolex

To calculate the communication cost, we use the average size of the models sent by all the participating clients per round as the metric. To calculate the computation overhead, we calculate the FLOPs and numbers of parameters in the models of all the participating clients per round and take the average as the metric. To put these metrics in context, we also calculate the upper and lower bounds of the communication cost and computation overhead (i.e., all the clients were using the same largest model and smallest model, respectively).

Table 7 lists the results. As shown, compared to the upper bound, `FedRolex` significantly reduces the communication cost and computation overhead while being able to achieve comparable model accuracy. Compared to the lower bound, although `FedRolex` has higher communication cost and computation overhead, the model accuracy achieved is much higher than the lower bound. These results indicate that `FedRolex` is able to achieve comparable high model accuracy as the upper bound with much less communication cost and computation overhead.

Table 7: Computation and communication costs of FedRolex compared to upper and lower bounds represented by homogeneous settings with largest and smallest models respectively.

| | Homogeneous (largest) | FedRolex | Homogeneous (smallest) |
|---|---|---|---|
| Average Number of Parameters per Client (Million) | 11.1722 | 2.9781232 | 0.04451 |
| Average FLOPs per Client (Million) | 557.656 | 149.048384 | 2.41318 |
| Average Model Size per Client (MB) | 42.62 | 11.36 | 0.17 |

### A.7 Experimental Setup Details

**Experimental Setup Details for Table 3.** The experimental setup for PT-based methods is listed in Table 8. The experimental setup for model-homogeneous baselines was slightly different from the PT-based methods and hence is listed separately in Table 9.

**Experimental Setup Details for Figure 3.** The experimental setup details are tabulated in Tables 10 and 11.

**Experimental Setup Details for Figure 4.** The experimental setup details are tabulated in Table 12.

Table 8: Experimental setup details of PT-based methods in Table 3 on CIFAR-10, CIFAR-100 and Stack Overflow.

| | | CIFAR-10 | CIFAR-100 | Stack Overflow |
|---|---|---|---|---|
| Local Epoch | | 1 | 1 | 1 |
| Cohort SIze | | 10 | 10 | 200 |
| Batch Size | | 10 | 24 | 24 |
| Initial Learning Rate | | 2.00E-04 | 1.00E-04 | 2.00E-04 |
| Decay Schedule | High Data Heterogeneity | 800, 1500 | 1000, 1500 | 600, 800 |
| | Low Data Heterogeneity | 800, 1250 | 1000, 1500 | |
| Decay Factor | | 0.1 | 0.1 | 0.1 |
| Communication Rounds | High Data Heterogeneity | 2500 | 3500 | 1200 |
| | Low Data Heterogeneity | 2000 | 3500 | |
| Optimizer | | SGD | SGD | SGD |
| Momentum | | 0.9 | 0.9 | 0.9 |
| Weight Decay | | 5.00E-04 | 5.00E-04 | 5.00E-04 |

Table 9: Experimental setup details of model-homogeneous baselines in Table 3 on CIFAR-10 and CIFAR-100 and Stack Overflow.

| | | CIFAR-10 | CIFAR-100 | Stack Overflow |
|---|---|---|---|---|
| Local Epoch | | 1 | 1 | 1 |
| Cohort Size | | 10 | 10 | 200 |
| Batch Size | | 10 | 24 | 24 |
| Initial Learning Rate | | 2.00E-04 | 1.00E-04 | 2.00E-04 |
| Decay Schedule | High Data Heterogeneity | 500, 1000 | 1000, 1500 | 300 |
| | Low Data Heterogeneity | 500, 1000 | 1000, 1500 | |
| Decay Factor | | 0.1 | 0.1 | 0.1 |
| Communication Rounds~ | High Data Heterogeneity | 1250 | 3500 | 1000 |
| | Low Data Heterogeneity | 1500 | 3500 | |
| Optimizer | | SGD | SGD | SGD |
| Momentum | | 0.9 | 0.9 | 0.9 |
| Weight Decay | | 5.00E-04 | 5.00E-04 | 5.00E-04 |

Table 10: Experimental setup for results shown in Figure 3. $\rho$ between $0.0$ and $0.5$ in $0.1$ increments.

| Dataset | | $\rho$ | 0.0 | 0.1 | 0.2 | 0.3 | 0.4 |
|---|---|---|---|---|---|---|---|
| CIFAR-10 | High Heterogeneity | Decay Schedule | 500, 1000 | 500, 1000 | 500, 1000 | 700, 1200 | 700, 1200 |
| | | Communication Rounds | 1250 | 1250 | 1250 | 1500 | 1500 |
| | Low Heterogeneity | Decay Schedule | 500, 1000 | 500, 1000 | 500, 1000 | 700, 1200 | 700, 1200 |
| | | Communication Rounds | 1250 | 1250 | 1250 | 1500 | 1500 |
| CIFAR-100 | High Heterogeneity | Decay Schedule | 1000, 1500 | 1000, 1500 | 1000, 1500 | 1000, 1500 | 1000, 1500 |
| | | Communication Rounds | 2000 | 2000 | 2000 | 2000 | 2000 |
| | Low Heterogeneity | Decay Schedule | 1000, 1500 | 1000, 1500 | 1000, 1500 | 1000, 1500 | 1000, 1500 |
| | | Communication Rounds | 2000 | 2000 | 2000 | 2000 | 2000 |
| Stack Overflow | High Heterogeneity | Decay Schedule | 800 | 800 | 800 | 800 | 800 |
| | | Communication Rounds | 1500 | 1500 | 1500 | 1500 | 1500 |
| | Low Heterogeneity | Decay Schedule | 800 | 800 | 800 | 800 | 800 |
| | | Communication Rounds | 1500 | 1500 | 1500 | 1500 | 1500 |

## A.8 Algorithm Pseudocodes

The pseudocodes for HeteroFL and Federated Dropout are given in Algorithms 2 and 3 respectively. Their differences from `FedRolex` are marked using blue color.

Table 11: Experimental setup for results shown in Figure 3. $\rho$ between 0.5 and 1.0 in 0.1 increments.

| Dataset | | $\rho$ | 0.6 | 0.7 | 0.8 | 0.9 | 1.0 |
|---|---|---|---|---|---|---|---|
| CIFAR-10 | High Heterogeneity | Decay Schedule | 700, 1200 | 700, 1200 | 500, 1000 | 500, 1000 | 500, 1000 |
| | | Communication Rounds | 1500 | 1500 | 1250 | 1250 | 1250 |
| | Low Heterogeneity | Decay Schedule | 700, 1200 | 700, 1200 | 500, 1000 | 500, 1000 | 500, 1000 |
| | | Communication Rounds | 1500 | 1500 | 1250 | 1250 | 1250 |
| CIFAR-100 | High Heterogeneity | Decay Schedule | 1000, 1500 | 1000, 1500 | 1000, 1500 | 1000, 1500 | 1000, 1500 |
| | | Communication Rounds | 2000 | 2000 | 2000 | 2000 | 2000 |
| | Low Heterogeneity | Decay Schedule | 1000, 1500 | 1000, 1500 | 1000, 1500 | 1000, 1500 | 1000, 1500 |
| | | Communication Rounds | 2000 | 2000 | 2000 | 2000 | 2000 |
| Stack Overflow | High Heterogeneity | Decay Schedule | 800 | 800 | 800 | 800 | 800 |
| | | Communication Rounds | 1500 | 1500 | 1500 | 1500 | 1500 |
| | Low Heterogeneity | Decay Schedule | 800 | 800 | 800 | 800 | 800 |
| | | Communication Rounds | 1500 | 1500 | 1500 | 1500 | 1500 |

Table 12: Experimental setup for results shown in Figure 4

| Dataset | | $\gamma$ | 2 | 4 | 8 | 16 |
|---|---|---|---|---|---|---|
| CIFAR-10 | High Heterogeneity | Decay Schedule | 800, 1200 | 800, 1200 | 800, 1200 | 800, 1200 |
| | | Communication Rounds | 1500 | 1500 | 1500 | 1500 |
| | Low Heterogeneity | Decay Schedule | 800, 1200 | 800, 1200 | 800, 1200 | 800, 1200 |
| | | Communication Rounds | 1500 | 1500 | 1500 | 1500 |
| CIFAR-100 | High Heterogeneity | Decay Schedule | 800, 1200 | 800, 1200 | 800, 1200 | 800, 1200 |
| | | Communication Rounds | 1500 | 1500 | 1500 | 1500 |
| | Low Heterogeneity | Decay Schedule | 800, 1200 | 800, 1200 | 800, 1200 | 800, 1200 |
| | | Communication Rounds | 1500 | 1500 | 1500 | 1500 |
| Stack Overflow | High Heterogeneity | Decay Schedule | 800 | 800 | 800 | 800 |
| | | Communication Rounds | 1500 | 1500 | 1500 | 1500 |
| | Low Heterogeneity | Decay Schedule | 800 | 800 | 800 | 800 |
| | | Communication Rounds | 1500 | 1500 | 1500 | 1500 |

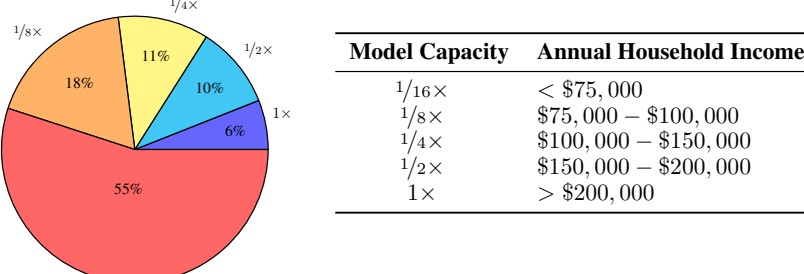

| Model Capacity | Annual Household Income |
|---|---|
| $1/16\times$ | $< \$75,000$ |
| $1/8\times$ | $\$75,000 - \$100,000$ |
| $1/4\times$ | $\$100,000 - \$150,000$ |
| $1/2\times$ | $\$150,000 - \$200,000$ |
| $1\times$ | $> \$200,000$ |

Figure 7: Mapping between real-world annual household income and model capacity.

Table 13: Experimental setup for Table 4 for CIFAR-10, CIFAR-100 and Stack Overflow.

| | | CIFAR-10 | CIFAR-100 | Stack Overflow |
|---|---|---|---|---|
| Local Epoch | | 1 | 1 | 1 |
| Cohort SIze | | 10 | 10 | 200 |
| Batch Size | | 10 | 24 | 24 |
| Initial Learning Rate | | 2.00E-04 | 1.00E-04 | 2.00E-04 |
| Decay Schedule | High Heterogeneity | 800, 1500 | 1000, 1500 | 600, 800 |
| | Low Heterogeneity | 800, 1250 | 1000, 1500 | |
| Decay Factor | | 0.1 | 0.1 | 0.1 |
| Communication Rounds | High Heterogeneity | 2500 | 3500 | 1200 |
| | Low Heterogeneity | 2000 | 3500 | |
| Optimizer | | SGD | SGD | SGD |
| Momentum | | 0.9 | 0.9 | 0.9 |
| Weight Decay | | 5.00E-04 | 5.00E-04 | 5.00E-04 |

---

**Algorithm 2: HeteroFL**

1 Initialization ; $\theta^{(0)}, \mathcal{N}$
  **Input** : $D_n \ \beta_n \ \forall n \in \mathcal{N}$,
  **Output** : $\theta^J$
2 Server Executes
3 **for** $j \leftarrow 0$ **to** $J - 1$ **do**
4    Sample subset $\mathcal{M}$ from $\mathcal{N}$
5    Broadcast $\theta^{(j)}_{m,[i \ ; \ 0,1, \ \ldots \ \lfloor \beta_n K_i \rfloor - 1]} \forall i$ and $m \in \mathcal{M}$
6    **for** *each client* $m \in \mathcal{M}$ **do**
7      clientStep($\theta^{(j)}_m, D_m$)
8    **end**
9    Aggregate $\theta^{(j+1)}_{[i,k]}$ according to Equation (10)
10 **end**
11 **Subroutine** clientStep($\theta^{(j)}_n, D_n$)
12    $m_n \longleftarrow len(D_n)$
13    **for** $k \leftarrow 0$ **to** $m_n$ **do**
14      $\theta_n \longleftarrow \theta_n - \eta \nabla l(\theta_n; d_{n,k})$
15    **end**
16    return $\theta_n$

---

**Algorithm 3: Federated Dropout**

1 Initialization ; $\theta^{(0)}, \mathcal{N}$
  **Input** : $D_n \ \beta_n \ \forall n \in \mathcal{N}$,
  **Output** : $\theta^J$
2 Server Executes
3 **for** $j \leftarrow 0$ **to** $J - 1$ **do**
4    Sample subset $\mathcal{M}$ from $\mathcal{N}$
5    Broadcast $\theta^{(j)}_{m,[i \ ; \ k_1,\ldots,k_{\lfloor \beta_n K_i \rfloor}]} \forall i$ and $m \in \mathcal{M}$
6    **for** *each client* $m \in \mathcal{M}$ **do**
7      clientStep($\theta^{(j)}_m, D_m$)
8    **end**
9    Aggregate $\theta^{(j+1)}_{[i,k]}$ according to Equation (10)
10 **end**
11 **Subroutine** clientStep($\theta^{(j)}_n, D_n$)
12    $m_n \longleftarrow len(D_n)$
13    **for** $k \leftarrow 0$ **to** $m_n$ **do**
14      $\theta_n \longleftarrow \theta_n - \eta \nabla l(\theta_n; d_{n,k})$
15    **end**
16    return $\theta_n$