# OpenReview forum: "FedRolex: Model-Heterogeneous Federated Learning with Rolling Sub-Model Extraction"
_NeurIPS.cc/2022/Conference — NeurIPS 2022 Accept_

### Official Review · Reviewer_ic9g · 2022-07-11

**Rating:** 4
**Confidence:** 4
**Soundness:** 2 fair
**Presentation:** 3 good
**Contribution:** 2 fair

**Summary:**

In this paper, the authors propose to sample the sub-models in a rolling way and find the simple method improves the performance of the global model compared with baselines.


**Questions:**

* KD-based baselines perform much worse than PT-based methods on CIFAR-100, could the authors provide the details of the chosen proxy data?
* Though sampling the sub-network in a rolling way outperforms baselines in the given experimental settings, it is not clear whether FedRolex outperforms baselines with a higher client participation rate.


**Limitations:**

This paper does not have a limitation subsection. But I don't think the paper will have potential negative societal impacts.

**Strengths And Weaknesses:**

Strengths:
* Overall, the paper provides an interesting perspective for understanding the limitations of HeteroFL and Federated Dropout.
* The authors not only evaluate the PT-based methods but also compare **FedRolex** with KD-based methods.
* In the appendix, the authors give statistical analysis to further analyze the proposed method.

Weakness:
* Compared with Federated Dropout and HeteroFL, FedRolex only modifies the sampling method for Federated Dropout. The idea is not novel enough.
* The experimental results do not list the performance of the simplest model in FedRolex. For the heterogeneous methods, the performance of each group should be listed.
* The authors only provide the statistical analysis to prove that the index is chosen equally randomly, the convergence analysis is not provided.
* The authors should provide the communication cost and computation overhead to reach the target accuracy.

---

> ### Author Response · Authors · 2022-07-30
> **Response to Reviewer ic9g (1/6)**
>
> We thank the reviewer for the thoughtful review. Here are our responses.
>
> ```(1/6) Compared with Federated Dropout and HeteroFL, FedRolex only modifies the sampling method for Federated Dropout. The idea is not novel enough.```
>
> We understand the reviewer's perspective as the idea presented in the main paper is focused on the simple rolling-based subnet extraction technique. In fact, we have introduced 2 other techniques including (1) client weighing, and (2) overlapping kernel extraction besides rolling-based subnet extraction. The reason we only presented the rolling-based subnet extraction technique in the main paper is based on the findings of our rigorous ablation studies: we found that the core idea of using the rolling-based subnet extraction is consistently outperforming SOTA baselines across multiple datasets and models. In contrast, as we have shown in our Appendix, our ablation studies on client weighing (Table 4), and overlapping kernel extraction (Figure 5) show that these techniques are not the main contributors to why we beat the SOTA baselines. We decided to not include those techniques in the main paper with the intention of not wanting to misguide the readers and instead pinpointing the most important technique that is consistently effective. We argue that despite the simplicity, we showed that it has a high impact in the sense that it allows the model to achieve accuracy close to homogeneous models.

---

> ### Author Response · Authors · 2022-07-31
> **Response to Reviewer ic9g (2/6)**
>
> ```(2/6) The experimental results do not list the performance of the simplest model in FedRolex. For the heterogeneous methods, the performance of each group should be listed.```
>
> Our goal in this paper is to train a large global model using a federation of clients with heterogeneous on-device resources. The models trained on the clients are parts of the large global model (i.e., submodels). Therefore, the accuracies of the submodels do not reflect the accuracy of the global model. This is why our work as well as  the other PT-based SOTA works only list the accuracies of the global model.
>
> Determining what models to deploy onto each heterogeneous client AFTER the global model has finished training during the federated learning process is a separate task. This can be achieved by a few techniques such as compressing/quantizing/KD the global model into smaller models to fit the resources of the target client. However, this is not the focus of this work and is a separate study in itself. We will pursue it in our future work.

---

> ### Author Response · Authors · 2022-07-31
> **Response to Reviewer ic9g (3/6)**
>
> ```(3/6) The authors only provide the statistical analysis to prove that the index is chosen equally randomly, the convergence analysis is not provided.```
>
> Similar to SOTA works (PT-based methods [1] [2] [3] and KD-based methods [4] [5] [6] [7]) we cite and compare against in this paper, we focus on the empirical study of the model-heterogeneity problem in federated learning. We admit that the full convergence analysis is actually not trivial, and instead, we provided statistical analysis to help readers understand why our method is better than other PT-based methods. We will make it clear in the revision to emphasize that our paper’s main contribution is from the empirical side, and state that the lack of a full convergence analysis is a limitation. We will pursue it in our future work.
>
> [1] Caldas, Sebastian, et al. "Expanding the reach of federated learning by reducing client resource requirements." arXiv preprint arXiv:1812.07210 (2018).
>
> [2]  Diao, Enmao, Jie Ding, and Vahid Tarokh. "HeteroFL: Computation and communication efficient federated learning for heterogeneous clients." arXiv preprint arXiv:2010.01264 (2020).
>
> [3] Horvath, Samuel, et al. "Fjord: Fair and accurate federated learning under heterogeneous targets with ordered dropout." Advances in Neural Information Processing Systems 34 (2021): 12876-12889.
>
> [4] Cho, Yae Jee, et al. "Heterogeneous Ensemble Knowledge Transfer for Training Large Models in Federated Learning." arXiv preprint arXiv:2204.12703 (2022).
>
> [5] He, Chaoyang, Murali Annavaram, and Salman Avestimehr. "Group knowledge transfer: Federated learning of large CNNs at the edge." Advances in Neural Information Processing Systems 33 (2020): 14068-14080.
>
> [6] Lin, Tao, et al. "Ensemble distillation for robust model fusion in federated learning." Advances in Neural Information Processing Systems 33 (2020): 2351-2363.
>
> [7] Itahara, Sohei, et al. "Distillation-based semi-supervised federated learning for communication-efficient collaborative training with non-iid private data." arXiv preprint arXiv:2008.06180 (2020).

---

> ### Author Response · Authors · 2022-07-31
> **Response to Reviewer ic9g (4/6)**
>
>  ```(4/6) The authors should provide the communication cost and computation overhead to reach the target accuracy.```
>
> To calculate the communication cost, we use the average size of the models sent by all the participating clients per round as the metric. Similarly, to calculate the computation overhead, we calculate the FLOPs and numbers of parameters in the models of all the participating clients per round and take the average as the metric.
>
> To put those calculated numbers into context, we also calculate the upper and lower bounds of the communication cost and computation overhead (i.e., all the clients were using the same model (largest and smallest respectively)). The results are listed in the table below.
> |                                                       | **Homogeneous(largest)** | **FedRolex** | **Homogeneous(smallest)** |
> | :---------------------------------------------------- | :----------------------- | :------------------- | :------------------------ |
> | **Average Number of Parameters per client (Million)** | 11\.1722                 | 2\.9781232           | 0\.04451                  |
> | **Average FLOPs per client (Million)**                | 557\.656                 | 149\.048384          | 2\.41318                  |
> | **Average model size per client (MB)**                | 42\.62                   | 11\.36               | 0\.17                     |
>
> As shown, we can see that compared to the upper bound, FedRolex significantly reduces the communication cost and computation overhead, while being able to achieve comparable model accuracy (see Table 2 in the submitted paper). Compared to the lower bound, although FedRolex has higher communication cost and computation overhead, the model accuracy achieved is much higher than the lower bound (see Table 2 in the submitted paper). Based on these results, we can conclude that FedRolex is able to achieve comparable high model accuracy as the upper bound with much less communication cost and computation overhead.
>
> We will add the results of this experiment in the Appendix.

---

> ### Author Response · Authors · 2022-07-31
> **Response to Reviewer ic9g (5/6)**
>
> ``` (5/6) KD-based baselines perform much worse than PT-based methods on CIFAR-100, could the authors provide the details of the chosen proxy data?```
>
> To clarify, since our method is PT-based other than KD-based, we directly pulled the performance of KD-based baselines from their original papers. For PT-based baselines, we duplicated them at our side following the codebases provided by the original papers.
>
> To answer the reviewer’s question, we quote the original KD-based paper [1] which provides the details of the chosen proxy data: “The public dataset is generated by applying a different data transformation to the data samples (non-overlapping with either the training or test dataset) to further differentiate it from the training dataset. For all datasets,non-overlapping users’ data samples are used.”
>
> [1] Cho, Yae Jee, et al. "Heterogeneous Ensemble Knowledge Transfer for Training Large Models in Federated Learning." arXiv preprint arXiv:2204.12703 (2022).

---

> ### Author Response · Authors · 2022-07-31
> **Response to Reviewer ic9g (6/6)**
>
> ```(6/6) Though sampling the sub-network in a rolling way outperforms baselines in the given experimental settings, it is not clear whether FedRolex outperforms baselines with a higher client participation rate.```
>
> In our paper, we used a 10% client participation rate by following works that are relevant to ours [1, 2, 3, 4]. To answer the reviewer’s question and examine the effect of client participation rate, we ran additional experiments with both lower (5%) and higher (20%) client participation rates using CIFAR10 as an example for FedRolex and both HeteroFL and Federated Dropout baselines. The results are summarized in the table below.
> |         |                   | Sample Rate       |                  |                    |
> | :------ | :---------------- | :---------------- | :--------------- | :----------------- |
> |         |                   | 5%                | 10%              | 20%                |
> | Cifar10 | HeteroFL          | 48\.43 (+/- 1.78) | 63\.90 (+/-2.74) | 65\.07 (+/- 2.17)  |
> |         | Federated Dropout | 42\.06 (+/- 1.29) | 46\.64 (+/-3.05) | 55\.20 (+/- 4.64)  |
> |         | FedRolex          | 57\.90 (+/- 2.72) | 69\.44 (+/-1.50) | 71\.85 ( +/- 1.22) |
>
> As shown, we can see that FedRolex consistently outperforms both Federated Dropout baselines under 5%, 10% and 20% client participation rates.
>
> We will add the results of this experiment in the Appendix.
>
> [1]  Diao, Enmao, Jie Ding, and Vahid Tarokh. "HeteroFL: Computation and communication efficient federated learning for heterogeneous clients." arXiv preprint arXiv:2010.01264 (2020).
>
> [2] Horvath, Samuel, et al. "Fjord: Fair and accurate federated learning under heterogeneous targets with ordered dropout." Advances in Neural Information Processing Systems 34 (2021): 12876-12889.
>
> [3] He, Chaoyang, Murali Annavaram, and Salman Avestimehr. "Group knowledge transfer: Federated learning of large CNNs at the edge." Advances in Neural Information Processing Systems 33 (2020): 14068-14080.
>
> [4] Reddi, Sashank, et al. "Adaptive federated optimization." arXiv preprint arXiv:2003.00295 (2020).

---

> ### Author Response · Authors · 2022-08-07
> **Thank you Reviewer ic9g**
>
> Dear Reviewer ic9g,
>
> We want to thank you for taking the time to review our paper and provide valuable comments to help us clarify and improve our work. Feel free to let us know if we have answered your questions and addressed your concerns. We really appreciate this opportunity to communicate with you to improve our work. Thanks again for the time and valuable comments. We really appreciate it.

---

### Official Review · Reviewer_2zgM · 2022-07-11

**Rating:** 8
**Confidence:** 5
**Soundness:** 4 excellent
**Presentation:** 4 excellent
**Contribution:** 4 excellent

**Summary:**

In this paper, the authors target an important problem in federated learning (FL) and propose a simple partial training (PT)-based technique called FedRolex to enable model-heterogeneous FL. Compared to existing methods, the core innovation in FedRolex is a rolling sub-model extraction scheme, where the sub-model is extracted from the global model using a rolling window that advances in each communication round. As such, parameters of the global model are evenly trained other than trained either randomly or statically in existing methods. This simple innovation has been demonstrated empirically to be the key factor that contributes to the superior performance of FedRolex over state-of-the-art methods.

**Questions:**

- Can the proposed technique generalize to other data types such as languages and the corresponding language models?
- Can the proposed technique generalize to a larger cohort size which is more practical in real-world deployments?


**Limitations:**

The authors discussed the limitations in section 5.

**Strengths And Weaknesses:**

Strengths:
- The review of the existing literature is solid. Table 1 summarizes the key differences between the proposed technique and existing works across a number of important dimensions. This is very helpful to understand the contributions of the paper.
- The proposed technique is simple yet powerful. Such simplicity helps the readers to factor out tricks that are not essential.
- The experiments are designed with careful thought and the conclusions drawn from the experimental results are quite exciting, especially enhancing the inclusiveness of FL in real-world distribution.

Weaknesses:
- The experiments are limited to vision datasets (i.e., CIFAR-10, CIFAR-10). It is not clear whether the proposed technique can generalize to other data types such as languages.
- The cohort size considered in the experiments is quite small. In practice, a much larger cohort size is normally incorporated in cross-device FL.

---

> ### Author Response · Authors · 2022-08-02
> **Response to Reviewer 2zgM (1/2)**
>
> We thank the reviewer for the thoughtful review. Here are our responses.
>
> (1/2) ```The experiments are limited to vision datasets (i.e., CIFAR-10, CIFAR-10). It is not clear whether the proposed technique can generalize to other data types such as languages.```
>
> ```Can the proposed technique generalize to other data types such as languages and the corresponding language models?```
>
>
> Our proposed method can indeed train SOTA models beyond CNN-based models such as Transformers. To demonstrate this, we applied our method to the Transformer model used for federated learning introduced in [1]. Specifically, the Transformer model includes 3 layers; the dimension of the token embeddings was 128; the hidden dimension of the feed-forward network (FFN) block is 2048; 8 heads were used for the multi-head attention, where each head is based on 12-dimensional (query, key, value) vectors; ReLU activation was used and the dropout rate was set to 0.1. To extract the submodels, we varied the width of the hidden layers in the Transformer heads to ½, ¼, ⅛, 1/16, and 1/32.
>
> In fact, the PT-based methods we used as baselines in our paper can be used to train the Transformer model. Therefore, we compare our method against PT-based baselines HeteroFL and Federated Dropout by training them using the Transformer model described above on the StackOverflow dataset, which consists of  342,477 clients. We followed [1] to sample 200 clients per round. We run the experiments three times with different seeds and report the average and std of the global accuracy. The results are listed in the table below.
>
> |     | **Method**           |     | **Global Accuracy**          |
> | :-- | :------------------- | :-- | :-------------------- |
> |     | Heterofl             |     | 27\.21 (+/- 0.12)     |
> |     | Federated Dropout    |     | 23\.46 (+/- 0.12)     |
> |     | *FedRolex*           |     | *29\.22 (+/- 0.24)* |
>
> As shown in the table, we can see that our proposed method outperforms both HeteroFL and Federated Dropout.
>
> We realize that in Figure 1 we visualized our method using convolution which has caused confusion. We will update Figure 1 where we plan to use more general neurons instead of convolution to indicate our method can also apply to Transformers.
>
> [1] Wang, Jianyu, et al. "A field guide to federated optimization." arXiv preprint arXiv:2107.06917 (2021).

---

> ### Author Response · Authors · 2022-08-02
> **Response to Reviewer 2zgM (2/2)**
>
> (2/2) ``` The cohort size considered in the experiments is quite small. In practice, a much larger cohort size is normally incorporated in cross-device FL.```
>
> ```Can the proposed technique generalize to a larger cohort size which is more practical in real-world deployments?```
>
> In our paper, we used a 10% client participation rate by following works that are relevant to ours [1, 2, 3, 4]. To answer the reviewer’s question and examine the effect of client participation rate, we ran additional experiments with both lower (5%) and higher (20%) client participation rates using CIFAR10 as an example for FedRolex and both HeteroFL and Federated Dropout baselines. The results are summarized in the table below.
> |         |                   | Sample Rate       |                  |                    |
> | :------ | :---------------- | :---------------- | :--------------- | :----------------- |
> |         |                   | 5%                | 10%              | 20%                |
> | Cifar10 | HeteroFL          | 48\.43 (+/- 1.78) | 63\.90 (+/-2.74) | 65\.07 (+/- 2.17)  |
> |         | Federated Dropout | 42\.06 (+/- 1.29) | 46\.64 (+/-3.05) | 55\.20 (+/- 4.64)  |
> |         | FedRolex          | 57\.90 (+/- 2.72) | 69\.44 (+/-1.50) | 71\.85 ( +/- 1.22) |
>
> As shown, we can see that FedRolex consistently outperforms both Federated Dropout baselines under 5%, 10% and 20% client participation rates.
>
> We will add the results of this experiment in the Appendix.
>
> [1]  Diao, Enmao, Jie Ding, and Vahid Tarokh. "HeteroFL: Computation and communication efficient federated learning for heterogeneous clients." arXiv preprint arXiv:2010.01264 (2020).
>
> [2] Horvath, Samuel, et al. "Fjord: Fair and accurate federated learning under heterogeneous targets with ordered dropout." Advances in Neural Information Processing Systems 34 (2021): 12876-12889.
>
> [3] He, Chaoyang, Murali Annavaram, and Salman Avestimehr. "Group knowledge transfer: Federated learning of large CNNs at the edge." Advances in Neural Information Processing Systems 33 (2020): 14068-14080.
>
> [4] Reddi, Sashank, et al. "Adaptive federated optimization." arXiv preprint arXiv:2003.00295 (2020).

---

> ### Author Response · Authors · 2022-08-07
> **Thank you Reviewer 2zgM**
>
> Dear Reviewer 2zgM,
>
> We want to thank you for taking the time to review our paper and provide valuable comments to help us clarify and improve our work. Feel free to let us know if we have answered your questions and addressed your concerns. We really appreciate this opportunity to communicate with you to improve our work. Thanks again for the time and valuable comments. We really appreciate it.

---

### Official Review · Reviewer_6GHV · 2022-07-11

**Rating:** 5
**Confidence:** 4
**Soundness:** 3 good
**Presentation:** 3 good
**Contribution:** 2 fair

**Summary:**

In this paper, the authors study the model heterogeneous problem in federated learning. The authors propose a simple but effective method, which adopts a rolling window to generate small sub-models for local training. Extensive on real-world datasets verify the effectiveness of the proposed method.

**Questions:**

Refer to the weakness.

**Limitations:**

Refer to the weakness.

**Strengths And Weaknesses:**

Strengths:
1. The model heterogeneity problem studied in this paper is important for the applications of federated learning in real-world scenarios.
2. The proposed method is simple to implement.
3. This paper is well-written and easy to follow.

Weakness:
1. The proposed method can only be used to train CNN-based models and is difficult to be applied to train other SOTA models, such as transformers.
2. The proposed method reduces computation costs for low-resource clients by only updating a small part of model parameters in a training round. The proposed method still needs different clients to train an intelligent model with the architecture. However, in some scenarios, it may be difficult for the low-resource clients to save and load the model that can be trained on the rich-resource clients. It seems that the proposed model cannot tackle such scenarios.
3. The computation resources in the rich-resource clients seem not be fully exploited by the proposed method.

---

> ### Author Response · Authors · 2022-07-30
> **Response to Reviewer 6GHV (1/3)**
>
> We thank the reviewer for the thoughtful review. Here are our responses.
>
> ```(1/3) The proposed method can only be used to train CNN-based models and is difficult to be applied to train other SOTA models, such as transformers.```
>
> Our proposed method can indeed train SOTA models beyond CNN-based models such as Transformers. To demonstrate this, we applied our method to the Transformer model used for federated learning introduced in [1]. Specifically, the Transformer model includes 3 layers; the dimension of the token embeddings was 128; the hidden dimension of the feed-forward network (FFN) block is 2048; 8 heads were used for the multi-head attention, where each head is based on 12-dimensional (query, key, value) vectors; ReLU activation was used and the dropout rate was set to 0.1. To extract the submodels, we varied the width of the hidden layers in the Transformer heads to ½, ¼, ⅛, 1/16, and 1/32.
>
> In fact, the PT-based methods we used as baselines in our paper can be used to train the Transformer model. Therefore, we compare our method against PT-based baselines HeteroFL and Federated Dropout by training them using the Transformer model described above on the StackOverflow dataset, which consists of  342,477 clients. We followed [1] to sample 200 clients per round. We run the experiments three times with different seeds and report the average and std of the global accuracy. The results are listed in the table below.
> |     | **Method**           |     | **Global Accuracy**          |
> | :-- | :------------------- | :-- | :-------------------- |
> |     | Heterofl             |     | 27\.21 (+/- 0.12)     |
> |     | Federated Dropout    |     | 23\.46 (+/- 0.12)     |
> |     | *FedRolex*           |     | *29\.22 (+/- 0.24)* |
>
> As shown in the table, we can see that our proposed method outperforms both HeteroFL and Federated Dropout.
>
> We realize that in Figure 1 we visualized our method using convolution which has caused confusion. We will update Figure 1 where we plan to use more general neurons instead of convolution to indicate our method can also apply to Transformers.
>
> [1] Wang, Jianyu, et al. "A field guide to federated optimization." arXiv preprint arXiv:2107.06917 (2021).

---

> ### Author Response · Authors · 2022-07-30
> **Response to Reviewer 6GHV (2/3)**
>
> ```(2/3) The proposed method reduces computation costs for low-resource clients by only updating a small part of model parameters in a training round. The proposed method still needs different clients to train an intelligent model with the architecture. However, in some scenarios, it may be difficult for the low-resource clients to save and load the model that can be trained on the rich-resource clients. It seems that the proposed model cannot tackle such scenarios.```
>
> Each client only needs to load the corresponding submodel that fits its local resource. Therefore, for low-resource clients, the submodels that they need to save and load can be much smaller than the global model.

---

> ### Author Response · Authors · 2022-07-30
> **Response to Reviewer 6GHV (3/3)**
>
> ```(3/3) The computation resources in the rich-resource clients seem not to be fully exploited by the proposed method.```
>
> Again, each client will load the corresponding submodel that fits its local resource. Therefore, for rich-resource clients, the submodels that they train can be as large as the global model. In fact, in Figure 3 in our paper, we showed that clients can use all their resources to train part of a large global model that it itself cannot fully load.

---

> ### Author Response · Authors · 2022-08-07
> **Thank you Reviewer 6GHV**
>
> Dear Reviewer 6GHV,
>
> We want to thank you for taking the time to review our paper and provide valuable comments to help us clarify and improve our work. Feel free to let us know if we have answered your questions and addressed your concerns. We really appreciate this opportunity to communicate with you to improve our work. Thanks again for the time and valuable comments. We really appreciate it.

---

> ### Comment · Reviewer_6GHV · 2022-08-08
> **Thank the authors for the response.**
>
> Thank the authors for your response, which addressed my major concerns. I have improved my ratting.

---

### Official Review · Reviewer_5GU1 · 2022-07-11

**Rating:** 8
**Confidence:** 4
**Soundness:** 3 good
**Presentation:** 4 excellent
**Contribution:** 4 excellent

**Summary:**

The authors proposed FedRolex, a model-heterogeneous federated learning framework that breaks the constraint of standard federated learning based on homogeneous model. The authors proposed a rolling technique to extract heterogeneous submodels from a shared global model, which has been shown to achieve much better performance in comparison with other model-heterogeneous federated learning methods such as Federated Dropout and HeteroFL in both high and low data heterogeneity levels.

**Questions:**

-	Would FedRolex work on modern large models such as Transformer?
-	Can the authors elaborate more on the concept of client drift to make the paper self-contained?


**Limitations:**

The authors discussed the limitations in section 5.

**Strengths And Weaknesses:**

Strengths:
- The authors did a good job in reviewing the existing works, especially the ones that are recently published.
- This is a timely and important topic to work on, and the authors have demonstrated that a simple submodel rotation technique can significantly bridge the gap between model homogeneous and model heterogeneous settings.
- The work has a potential to greatly improve the fairness of federated learning.

Weaknesses:
- The primary weakness of the work is that although the work claims that the proposed FedRolex could train large server models, the model studied in the current version is ResNet18. The work would be much more convincing if FedRolex could train modern large models such as Transformers.

---

> ### Author Response · Authors · 2022-08-02
> **Response to Reviewer 5GU1 (1/2)**
>
> We thank the reviewer for the thoughtful review. Here are our responses
>
> (1/2)
> ``` The primary weakness of the work is that although the work claims that the proposed FedRolex could train large server models, the model studied in the current version is ResNet18. The work would be much more convincing if FedRolex could train modern large models such as Transformers.```
>
> ```Would FedRolex work on modern large models such as Transformer?```
>
>
> Our proposed method can indeed train SOTA models beyond CNN-based models such as Transformers. To demonstrate this, we applied our method to the Transformer model used for federated learning introduced in [1]. Specifically, the Transformer model includes 3 layers; the dimension of the token embeddings was 128; the hidden dimension of the feed-forward network (FFN) block is 2048; 8 heads were used for the multi-head attention, where each head is based on 12-dimensional (query, key, value) vectors; ReLU activation was used and the dropout rate was set to 0.1. To extract the submodels, we varied the width of the hidden layers in the Transformer heads to ½, ¼, ⅛, 1/16, and 1/32.
>
> In fact, the PT-based methods we used as baselines in our paper can be used to train the Transformer model. Therefore, we compare our method against PT-based baselines HeteroFL and Federated Dropout by training them using the Transformer model described above on the StackOverflow dataset, which consists of  342,477 clients. We followed [1] to sample 200 clients per round. We run the experiments three times with different seeds and report the average and std of the global accuracy. The results are listed in the table below.
>
> |     | **Method**           |     | **Global Accuracy**          |
> | :-- | :------------------- | :-- | :-------------------- |
> |     | Heterofl             |     | 27\.21 (+/- 0.12)     |
> |     | Federated Dropout    |     | 23\.46 (+/- 0.12)     |
> |     | *FedRolex*           |     | *29\.22 (+/- 0.24)* |
>
> As shown in the table, we can see that our proposed method outperforms both HeteroFL and Federated Dropout.
>
> We realize that in Figure 1 we visualized our method using convolution which has caused confusion. We will update Figure 1 where we plan to use more general neurons instead of convolution to indicate our method can also apply to Transformers.
>
> [1] Wang, Jianyu, et al. "A field guide to federated optimization." arXiv preprint arXiv:2107.06917 (2021).

---

> > ### Comment · Reviewer_5GU1 · 2022-08-08
> > **Response to Authors**
> >
> > Dear Authors,
> >
> > Many thanks for your great efforts in the rebuttal, especially for the extra experiments. Hope all rebuttal results and discussion can be delivered in the revision. I will raise my score.
> >
> > Best wishes,
> > Reviewer 5GU1

---

> > > ### Author Response · Authors · 2022-08-10
> > > **Revision Updated**
> > >
> > > Dear Reviewer 5GU1,
> > >
> > > We have included the rebuttal results and discussion into the revision, and have uploaded the latest version. Thanks again for the time and valuable comments. We really appreciate it.

---

> ### Author Response · Authors · 2022-08-02
> **Response to Reviewer 5GU1 (2/2)**
>
> (2/2) ```Can the authors elaborate more on the concept of client drift to make the paper self-contained?```
>
> The term ‘Client Drift’ was mentioned in [1]. Here, it was used to describe the error in gradient estimation due to the differences in data distributions between clients. The error in gradient estimation (or model update) can arise for several reasons in federated learning. With multiple local updates, each client implements (stochastic) gradient update several times based on its own updated models. For model heterogeneous federated learning, client drift will occur not only due to the differences in data distributions between the clients' private datasets but also due to the inconsistency between the clients' individual model architectures.
>
> [1] Wang, Jianyu, et al. "A field guide to federated optimization." arXiv preprint arXiv:2107.06917 (2021).

---

> ### Author Response · Authors · 2022-08-07
> **Thank you Reviewer 5GU1**
>
> Dear Reviewer 5GU1,
>
> We want to thank you for taking the time to review our paper and provide valuable comments to help us clarify and improve our work. Feel free to let us know if we have answered your questions and addressed your concerns. We really appreciate this opportunity to communicate with you to improve our work. Thanks again for the time and valuable comments. We really appreciate it.

---

### Meta-Review · Area_Chair_LS2o · 2022-08-20

**Recommendation:** Accept
**Confidence:** Less certain

**Metareview:**

To better handle the case that each client is with heterogeneous device resources, this paper presents a model-heterogeneous federated learning algorithm FedRolex. FedRolex rolls the submodel in each federated iteration, in order to train the parameters of the global model on the global data distribution. Experimental results show FedRolex outperforms other model-heterogeneous baselines. Ablation studies on submodel rolling show it is an effective technique.

However, this paper suffers from several limitations. Firstly, it remains unclear why FedRolex can significantly outperform Federated Dropout and HeteroFL, since they are only different in sampling methods. Secondly, after federated learning, the low-end devices still can only use a sub-model. Will it benefit?

**Award:**

No

---

### Decision · Program_Chairs · 2022-09-14

Accept